# Comparison of functional-oil blend and anticoccidial antibiotics effects on performance and microbiota of broiler chickens challenged by coccidiosis

**Paula Gabriela da Silva Pires**[1☯], **Pedro Torres**[2☯], **Tatiany Aparecida Teixeira Soratto**[3☯],
**Vilmar Benetti Filho**[3☯], **Lucélia Hauptli**[2☯], **Glauber Wagner**[3☯], **Douglas Haese**[4☯], **Carolina D'ávila Pozzatti**[4☯], **Priscila de Oliveira Moraes**[1,2☯] *

**1** Advanced Poultry Gut Science, Florianópolis, Santa Catarina, Brazil, **2** Department of Animal Science and Rural Development, Universidade Federal de Santa Catarina, Florianópolis, Santa Catarina, Brazil, **3** Laboratory of Bioinformatics, Center of Biological Sciences, Universidade Federal de Santa Catarina, Florianópolis, Santa Catarina, Brazil, **4** Centro de Tecnologia Animal Ltda, Domingos Martins, ES, Brazil

☯ These authors contributed equally to this work.
* priscila.moraes@ufsc.br

**Data Availability Statement:** All relevant data are within the paper and its Supporting Information files.

## Abstract

This study aimed to compare the effects of different levels of cashew nutshell liquid (CNSL) and castor oil (CNSL–castor oil) with growth-promoting antibiotics associated with anticoccidials in broiler chickens challenged with coccidiosis. In this work, 2520 one-day-old male broiler chicks (Cobb) were randomly assigned to 84 pens, containing 30 birds each. The experimental design was completely randomized, with seven treatments: enramycin (8 ppm), virginiamycin (16.5 ppm), and tylosin (55 ppm); different doses of CNSL–castor oil (0.5, 0.75, and 1.00 kg/t); and a control diet (without additives). All treatments received semduramicin + nicarbazin (500 g/t; Aviax® Plus) from 0 to 28 d and monensin sodium (100 ppm; Elanco) from 29 to 35 days of age, when the feed was without antibiotics. The challenge was introduced at 14 days of age by inoculating broiler chickens with sporulated *Eimeria tenella*, *Eimeria acervulina*, and *Eimeria maxima* oocysts via oral gavage. In addition to performance parameters, intestinal contents were collected at 28 and 42 days of age for microbiota analysis by sequencing the 16s rRNA in V3 and V4 regions using the Illumina MiSeq platform. Taxonomy was assigned using the SILVA database (v. 138) with QIIME2 software (v. 2020.11). After one week of challenge, the broilers that received tylosin had a higher body weight gain (BWG) than those in the control group ($p < 0.05$), while the other treatments presented intermediate values. At 28 d, the BWG was lower for the control, CNSL–Castor oil 0.5 kg/t, enramycin, and virginiamycin treatments than that in the tylosin treatment. The inclusion of CNSL–Castor oil at concentrations of 0.75 and 1 kg/t acted as an intermediate treatment ($p < 0.05$). For alpha diversity, using the Shannon index, it was possible to observe the effect of age, with substantial diversity at 42 d. The Firmicutes phylum had the highest abundance, with values between 84.33% and 95.16% at 42 d. Tylosin showed better performance indices than other treatments. CNSL–castor oil treatments with concentrations of 0.75 and 1 kg/t showed similar results to those of enramycin and

**Funding:** The author(s) received no specific funding for this work.

**Competing interests:** The authors have declared that no competing interests exist.

virginiamycin. Furthermore, CNSL–castor oil acted as a modulator of intestinal microbiota, reducing the abundance of pathogenic bacteria.

## Introduction

Coccidiosis is an infection caused by *Eimeria* spp. that seriously affects poultry farming worldwide. Recent studies have estimated a worldwide loss of approximately £10.36 billion per year owing to prophylaxis and production losses in chickens due to coccidiosis [1]. The parasite affects the intestinal epithelium and immune response of chickens, reduces nutrient digestion, and consequently has a negative impact on the performance indices of broilers [2]. Additionally, coccidiosis disturbs the diversity and composition of the intestinal microbiota. For example, the increased leakage of plasma proteins into the lumen induced by the parasite provides substrate for the proliferation of *Clostridium perfringens* [3]. Although this bacterium is part of the normal microbiota of broiler cecum, when there is a significant increase in its proliferation in the small intestine, it can cause necrotic enteritis [2, 3].

Over the years, antibiotic growth promoters (AGP) have been used to control pathogens and preserve intestinal integrity and enhance production indicators for broilers. The performance improvement because of AGP is associated with the modification of the intestinal microbiota. AGP promotes a balance in the microbial population, as it reduces the number of toxin-producing microorganisms in the intestinal lumen [4], in addition to acting as an antibacterial and direct anti-inflammatory agent [5].

However, some countries have banned the use of AGPs because of the risks to human health caused by residues in animal products, as well as the possibility of inducing bacterial resistance [6]. In addition, a meta-analysis showed that the effectiveness of antibiotics as growth promoters was less evident in recently published studies than that in the 1980s [7]. Thus, the production response to AGP usage could be reduced with good production conditions, such as "hygienic facilities and balanced nutrition [7].

The initiative to reduce the use of antibiotics as growth promoters has stimulated research into alternative methods to simultaneously minimize the impact of parasites and act as growth promoters. The balance of intestinal microbiota is crucial for good animal performance, especially when facing sanitary challenges. In addition, Moraes et al. [8] and Vieira et al. [9] demonstrated that when receiving a commercial product consisting of a mixture of cashew nutshell liquid and castor oil (CNSL–castor oil), chickens challenged by coccidiosis presented a better balance in their microbiota, reducing pathogenic bacteria such as *Clostridium perfringens* and improving animal performance.

This study aimed to compare the effects of CNSL–castor oil with multiple growth-promoting antibiotics used as anticoccidials in broiler chickens challenged by coccidiosis.

## Material and methods

### Ethics statement

The Center for Animal Technology Ethics Committee (CTA) approved all procedures used in this experiment on the use of animals on 11/21/2019, under protocol number 001.57/19, following Law No. 11.794 of October 8, 2008, Decree No. 6.899 of July 15, 2009, and with the norms published by the National Council for the Control of Animal Experimentation (CONCEA). Throughout the experiment, all animals were monitored twice daily. Observations were carried out systematically by trained technicians who evaluated possible clinical signs of

coccidiosis, such as occurrence of mucus-like or bloody diarrhea, dehydration diagnosed by wrinkled skin or dull eyes, decreased appetite detected by a decrease in feed intake, occurrence of ruffled feathers, listlessness detected by a lack of bird activity, and stunted growth that was visibly lower than the group average. In cases of early evidence associated with possible risk or specific signs of severe suffering or distress, the plans called for the birds to be euthanized. However, there was no specific instance where euthanasia was required; the coccidiosis infection occurred at the planned level, causing only mild symptoms.

## Bird husbandry and experimental design

A total of 2520 one-day-old male chicks (Cobb 500) were obtained from a commercial hatchery and housed in two identical experimental rooms, with a total of 84 boxes at an initial density of 30 birds per box. The nutritional program consisted of two diets: initial phase (1–28 d) and final phase (28–42 d), based on the nutritional requirements recommended by the Brazilian Tables of Poultry and Swine [10]. The nutritional composition was the same for all treatments, varying only in the additives used (S1 Table). All treatments received semduramicin + nicarbazin (500 g/t; Aviax® Plus) from 0 to 28 d and monensin sodium (100 ppm; Elanco) from 29 to 35 days of age, when the feed was without antimicrobial components. The experimental period lasted 42 d, broilers were weighed weekly, feed intake was measured, and feed conversion was calculated [11].

The experimental design was completely randomized using 7 treatments with 12 replicates including 30 birds each. The following treatments were used: enramycin (8 ppm, MSD Animal Health), virginiamycin (16.5 ppm, Phibro Animal Health, Teaneck, NJ, USA), or tylosin (55 ppm, Elanco Animal Health, Greenfield, IN, US); different doses of CNSL–Castor oil (0.5, 0.75, and 1.00 kg/t); and a control treatment (no additives). All antimicrobial doses were used at sub-therapeutic levels according to the manufacturers' instructions for disease prevention or growth promotion.

## Challenge and sample collection

At 14 days of age, 1 mL sporulated oocysts of *E. tenella* ($10 \times 10^3$), *E. acervulina* ($200 \times 10^3$), and *E. maxima* ($80 \times 10^3$) were inoculated by gavage. Oocysts were acquired from the Animal Technology Center Ltd. (Espirito Santo, Brazil). After 7 and 14 d of oocyst inoculation (21 and 28 d of age), two birds with average weight from each replicate (168 birds per period) were euthanized by cervical dislocation, and the *Eimeria spp.* lesion score was evaluated. Lesions were ranked from 0 (absence of macroscopic lesions) to 4 (presence of severe macroscopic lesions), according to a previously described method [12].

For microbiota sequencing analysis, the cecum of two birds per replicate was collected, homogenized, pooled, and immediately stored at -20˚C, at 28 and 42 days of age, with a total of 12 samples per treatment.

## DNA extraction, PCR amplification, and sequencing

DNA was extracted using the QIAamp DNA Stool Mini (QIAGEN, Hilden, Germany) following the manufacturer's recommendations. The 16S rRNA V3/V4 region was amplified using primers 341F (5′–CCTACGGGRSGCAGCAG–3′) and 806R (5′–GGACTACHVGGGTWTCTAAT–3′) with Illumina adapters for sequencing. Amplification was performed in 35 cycles at an annealing temperature of 50˚C, and analysis was conducted in triplicate. Sequencing was performed on an Illumina MiSeq platform using the V2-500 kit with a 500 bp paired-end run.

## Microbial diversity assessment

The sequencing data were processed using QIIME2 (v. 2020.11) [13]. Low-quality sequences (Phred Quality < 25) were removed, sequencing errors were corrected, chimeras were removed, and amplicon sequence variants (ASVs) were identified using the DADA2 method, executed with default parameters and forward read sequences truncated at 300 bp and reverse truncated at 200 bp. ASVs with a frequency below 0.1% were removed. The taxonomy was assigned to ASVs using the naïve Bayes approach implemented using the *scikit* learn Python library with default parameters and the SILVA database (v. 138).

Relative abundance, as well as alpha (Chao-1, Shannon, and Simpson), and beta diversity indices, were calculated using the R program (v. 3.6.1) (https://www.R-project.org/) with the reshape2 (v. 1.4.3) [14] and phyloseq (v. 1.14.0) [15] packages. Beta diversity was estimated after normalizing the sequence number by randomly choosing the same number of sequences in each sample. After normalization, principal coordinate analysis (PCoA) was performed using the Bray-Curtis dissimilarity index in the vegan package (v. 2.4.1) [16].

## Statistical analysis

Statistical analyses were performed using SAS statistical software (version 9.0; SAS Inst. Inc., Cary, NC, United States). The data were subjected to analysis of variance using PROC GLM, with each box considered as an experimental unit. Differences (p < 0.05) were assessed using Tukey's multiple comparison test. Regression analysis was performed to check the effect of the CNSL–castor oil inclusion level. Injury scores were evaluated using the Student–Newman–Keuls (SNK) method. The microbiota indices were assessed using the Kruskal–Wallis test, and each treatment was compared pairwise using the Wilcoxon test corrected by the false discovery rate. Beta diversity distances were compared using the Adonis test.

## Results

### Performance data and lesion score

The performance responses, separated into periods including days 1–21, 21–28, 28–42, and 1–42, are listed in Table 1. At 21 d, the broilers that received tylosin presented a statistically significant increased weight gain compared with control animals (p < 0.05), and consequently, tylosin-treated birds had higher live weight and better feed conversion ratio (p < 0.05). The control group showed the poorest feed conversion (P < 0.05).

After 14 d of challenge by Eimeria spp., at 28 d of age, the animals that received tylosin had higher feed intake than broilers treated with enramycin (p < 0.05). The other treatments showed no significant differences. Animals treated with the antibiotic tylosin showed more significant weight gain than those treated with enramycin and the control. The other treatments exhibited intermediate behaviors. The live weight at 28 d was lower for the birds in the control, CNSL–Castor oil 0.5 kg/t, enramycin, and virginiamycin groups compared with those in the tylosin group; and CNSL–Castor oil 0.75 and 1 kg/t groups behaved as intermediate treatments (p < 0.05). Feed conversion was higher in control birds than that in other treatment groups (p < 0.05), except for CNSL–Castor oil (0.5 kg/t). In the last phase, from 28 to 42 d, the broilers showed greater weight gain when receiving tylosin and enramycin than the control broilers, CNSL–Castor oil 0.5 kg/t, or CNSL–Castor oil 0.75 kg/t groups (P < 0.05); animals that received other treatments showed intermediate effects.

Assessing the complete period of 1–42 d, the treatment with tylosin provided a greater weight gain than the control or any CNSL–Castor oil treatments. Treatment with

**Table 1. Live weight (LW), body weight gain (WG), feed intake (FI), and feed conversion ratio (FCR) of challenged broilers from 1 to 42 d of age.**

| Item | Additives | | | | | | | SEM[2] | p-value |
|---|---|---|---|---|---|---|---|---|---|
| | Control | CNSL-CO [1] 0.5 kg/t | CNSL-CO 0.75 kg/t | CNSL-CO 1.0 kg/t | Enramycin | Virginiamycin | Tylosin | | |
| LW (1D), g | 0.037 | 0.037 | 0.037 | 0.037 | 0.037 | 0.037 | 0.037 | 0.00007 | 0.476 |
| *Phase 1 to 21 days* | | | | | | | | | |
| LW, g | 773 a | 757 ab | 749 ab | 745 ab | 740 ab | 739 ab | 778 b | 0.00369 | 0.036 |
| BWG, g | 33.41 b | 34.3 ab | 33.91 ab | 33.72 ab | 33.51 ab | 33.45 ab | 35.28 a | 0.00018 | 0.035 |
| FI, g | 45.75 | 46.35 | 44.94 | 45.27 | 45.14 | 44.44 | 46.00 | 0.00025 | 0.420 |
| FCR, g/g | 1.370 a | 1.352 ab | 1.327 ab | 1.343 ab | 1.348 ab | 1.329 ab | 1.303 b | 0.00586 | 0.000 |
| *Phase 21 to 28 days* | | | | | | | | | |
| LW, g | 1242 b | 1289 b | 1295 ab | 1292 ab | 1260 b | 1283 b | 1355 a | 0.00610 | 0.000 |
| BWG, g | 72.28 b | 75.98 ab | 77.95 ab | 78.14 ab | 74.25 b | 77.70 ab | 82.49 a | 0.00070 | 0.003 |
| FI, g | 121.96 ab | 121.30 ab | 122.05 ab | 122.02 ab | 117.33 b | 120.77 ab | 125.78 a | 0.00061 | 0.021 |
| FCR, g/g | 1.696 a | 1.601 ab | 1.572 b | 1.567 b | 1.585 b | 1.559 b | 1.527 b | 0.01050 | 0.000 |
| *Phase 28 to 42 days* | | | | | | | | | |
| LW, g | 2548 c | 2620 bc | 2632 bc | 2640 bc | 2672 ab | 2656 abc | 2767 a | 0.01210 | 0.000 |
| BWG, g | 93.29 b | 95.13 b | 95.50 b | 96.30 ab | 100.84 a | 98.08 ab | 100.88 a | 0.00075 | 0.037 |
| FI, g | 179.79 | 177.87 | 179.13 | 178.18 | 177.31 | 177.89 | 176.54 | 0.00077 | 0.951 |
| FCR, g/g | 1.932 a | 1.872 ab | 1.879 ab | 1.861 ab | 1.759 b | 1.816 ab | 1.760 b | 0.01220 | 0.000 |
| *Phase 1 to 42 days* | | | | | | | | | |
| BWG, g | 59.78 c | 61.52 bc | 61.72 bc | 61.98 bc | 62.75 ab | 62.37 abc | 65.02 a | 0.00029 | 0.000 |
| FI, g | 103.13 | 102.68 | 102.52 | 102.65 | 101.23 | 101.65 | 102.8 | 0.00036 | 0.817 |
| FCR, g/g | 1.726 a | 1.669 ab | 1.660 b | 1.654 b | 1.613 bc | 1.630 bc | 1.583 c | 0.00697 | 0.000 |

Means with different letters differ statically by Tukey, on the row within the same variable. Data are expressed as means of the information collected in 360 broilers per treatment.

[1] CNSL-CO: CNSL-castor oil. Essential (US Patent N°. 8377,485; Oligo Basics Ind. Ltda., Cascavel, Paraná, Brazil).

[2] SEM: standard error of the mean.

enramycin and virginiamycin did not differ from other treatments (p > 0.05). Consequently, the pattern of live weight at 42 d was identical to that of weight gain. Feed intake did not differ between the treatments in the last phase or in the total period (p > 0.05). Tylosin showed comparatively better feed conversion, followed by CNSL–Castor oil levels of 0.75 and 1.0 kg/t. However, the CNSL–Castor oil 0.75 and 1.0 kg/t treatments did not significantly differ from treatments with enramycin and virginiamycin. In turn, neither antibiotic treatment differed from that with tylosin. As expected, the control treatment resulted in the lowest feed conversion.

In the first week after infection, broilers subjected to the control treatment had a higher lesion score for *E. acervulina* and *E. tenella* compared with other treatments (p < 0.05). At 28 d, there was no statistical difference in the injury score among the treatments (S1 Fig).

## Variations in alpha and beta diversity of microbiota

A total of 3,369,364 sequences with good quality were identified (20,055.74 ± 12,758.68 per sample). Samples 200924161005-1-1-1 (virginiamycin—28 d), 200924160996-1-1-1 (virginiamycin—28 d), and 200924160978-1-1-1 (CNSL–CO 1.00–28 d) showed low sequencing depths and were disregarded from subsequent analyses (S2 Table).

Alpha diversity was calculated using the Chao-1, Shannon, and Simpson indices. There was a trend of increasing diversity from 28 to 42 d when using the three indices; only the Simpson index presented significant results, indicating a higher time dominance at 42 d (Fig 1).

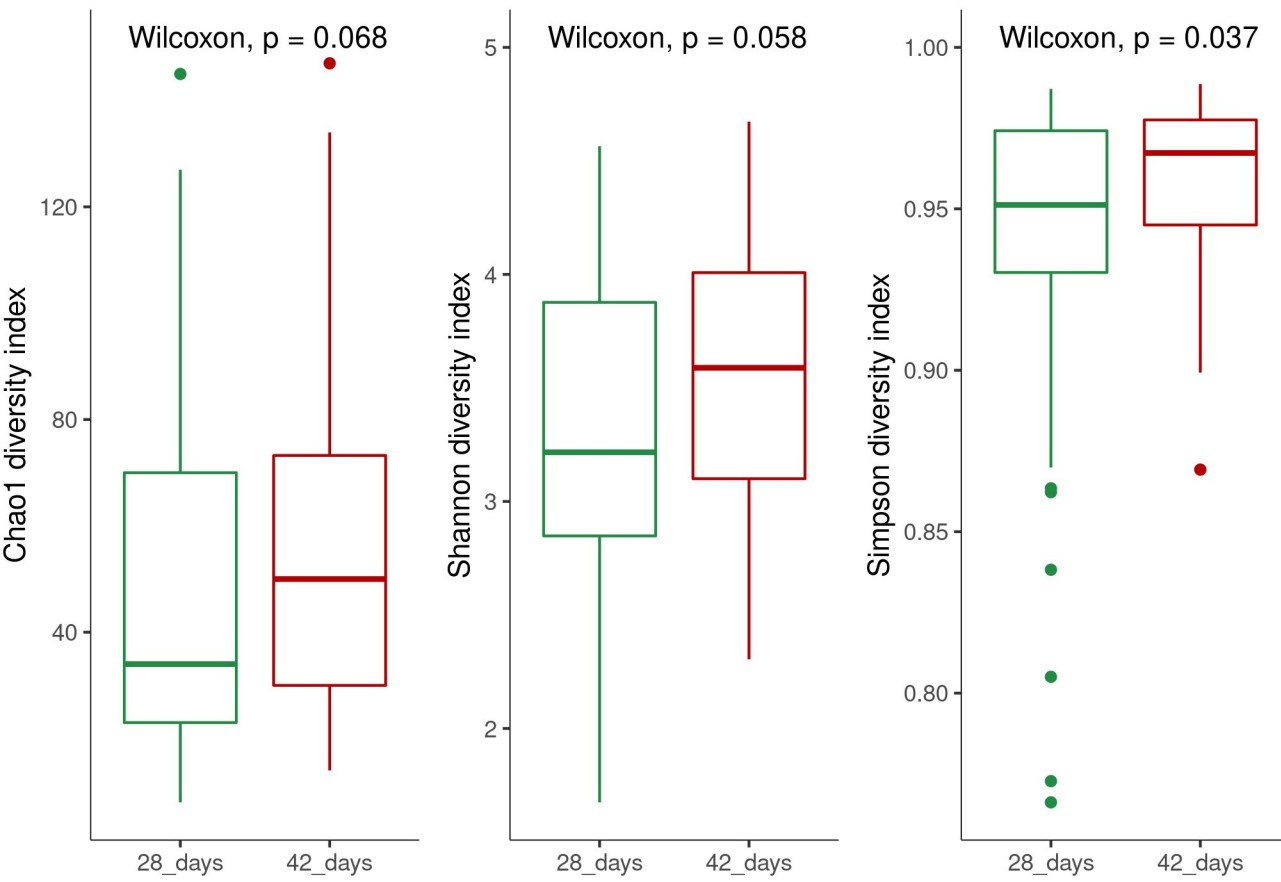

**Fig 1. Alpha diversity of broilers cecum samples at 28 and 42 d of trial.** The study tested seven feed additives: enramycin (8 ppm), virginiamycin (16.5 ppm), tylosin (55 ppm), CNSL–castor oil (CNSL-CO) at different doses (0.5, 0.75, and 1.00 kg/t), and the control diet (without additives). Comparison over time (28 and 42 d) using Chao-1, Shannon, and Simpson indices.

Comparing the alpha diversity-calculated Shannon index for each treatment at different times, we observed that for Control, CNSL–castor oil 0.5 kg/t, CNSL–castor oil 0.75 kg/t, and tylosin treatments, there was an increase in diversity from 28 to 42 d. The opposite was observed for virginiamycin, whereof the diversity at 28 d was greater than that at 42 d. No differences were observed with the additives enramycin and CNSL–castor oil 1.00 kg/t (Fig 2). The same trend was observed for indices (S2A and S2B Fig).

Comparing the different additives at each time point, we observed statistically significant differences between them (Kruskal–Wallis, P < 0.05). At 28 d, there were no significant differences compared with the control (Fig 3). At 42 d, the control differed from treatments with virginiamycin, CNSL–castor oil 1.00 kg/t, and enramycin (S3A and S3B Fig).

Bray–Curtis dissimilarity values were calculated to measure individual differences in taxonomic structure. To visualize the differences between the microbiome profiles, we performed PCoA for the Bray–Curtis dissimilarity matrix, which did not reveal a clear grouping pattern. All samples were scattered with overlapping ellipses at 28 and 42 d (Fig 4A). At day 42, the differences observed were statistically significant (Adonis, p < 0.05) (Table 2). Fig 4A shows the distribution of paired beta diversity values between treatments in each period studied; the distributions were not similar. Comparing each treatment at different times, we found that CNSL–castor oil 0.5 kg/t, enramycin, virginiamicin, and tylosin treatments differed statistically from 28 to 42 d (Adonis, p < 0.05) (Fig 4B and Table 2).

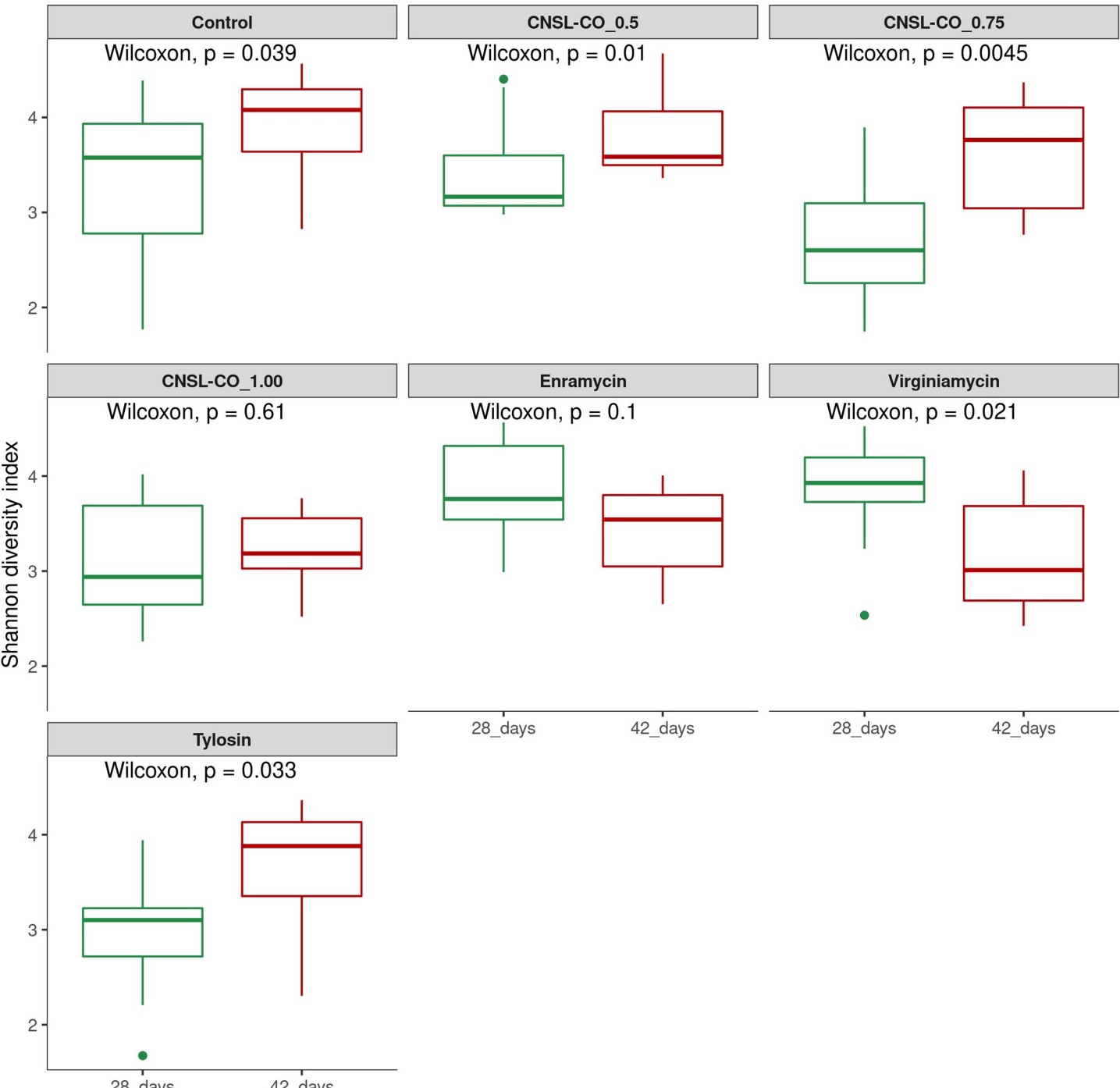

**Fig 2. Comparison of alpha diversity for each additive between time points using Shannon index of broilers cecum samples at 28 and 42 d of trial.** The study tested seven feed additives: enramycin (8 ppm), virginiamycin (16.5 ppm), tylosin (55 ppm), CNSL-castor oil (CNSL-CO) at different doses (0.5, 0.75, and 1.00 kg/t), and the control diet (without additives).

## Overall taxonomic composition

The processed sequences were assigned to 5180 ASVs. All sequences were classified into nine phyla, although four phyla were most common (> 1%): Firmicutes, Bacteroidetes,

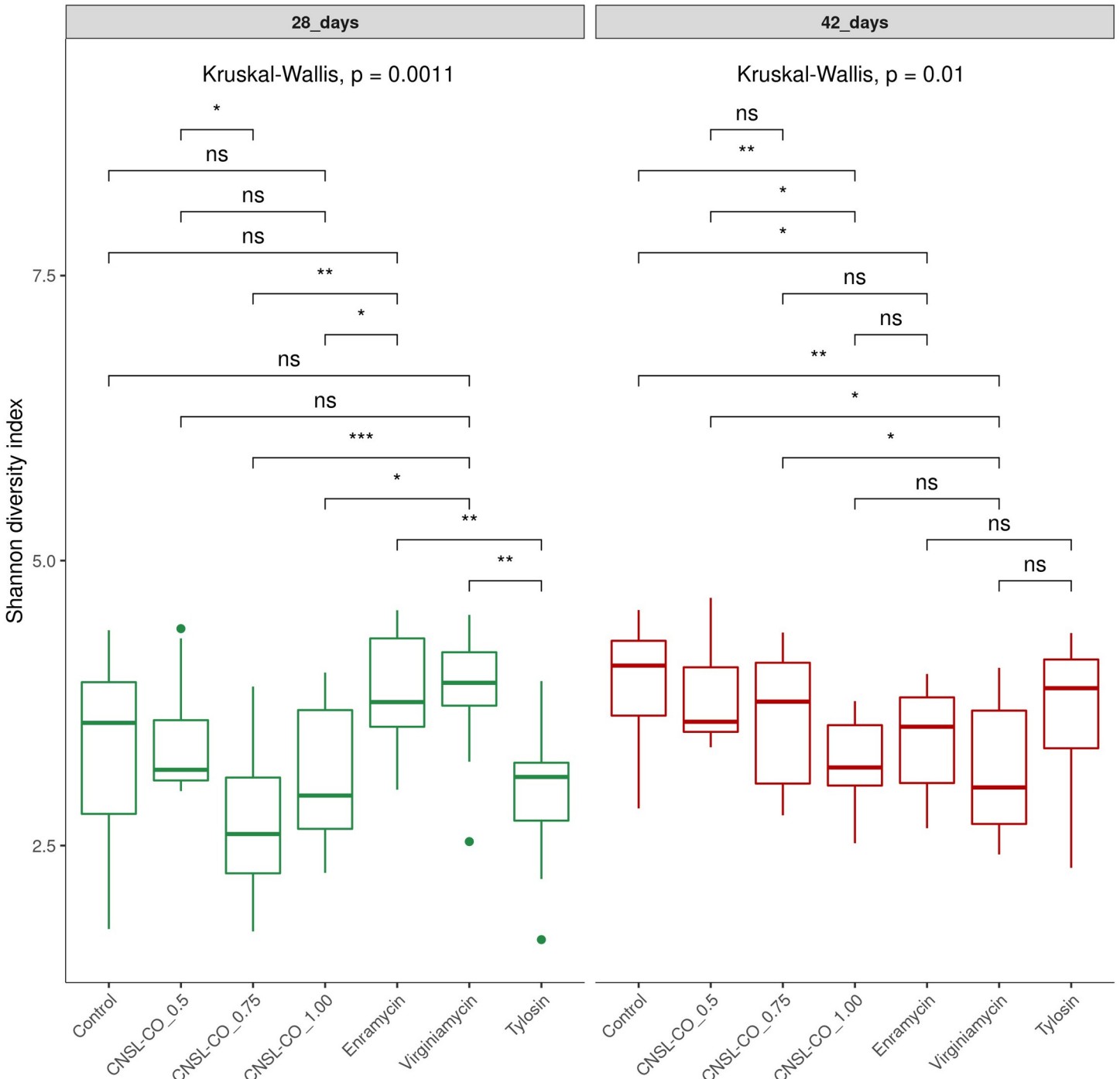

**Fig 3. Comparison of alpha diversity at each time point using Shannon index of broilers cecum samples at 28 and 42 d of trial.** The study tested seven feed additives: enramycin (8 ppm), virginiamycin (16.5 ppm), tylosin (55 ppm), CNSL-castor oil (CNSL-CO) at different doses (0.5, 0.75, and 1.00 kg/t), and the control diet (without additives). Probabilities and means with different letters differ statistically by Tukey: 0 '\*\*\*' 0.001 '\*\*' 0.01 '\*' 0.05.

Actinobacteria, and Proteobacteria. Firmicutes was the most abundant phylum in all treatments (> 84%) (Fig 5A). A complete list of the identified sequences (relative abundances) per treatment is provided in S3 Table. Of the 47 families identified, 19 had a relative

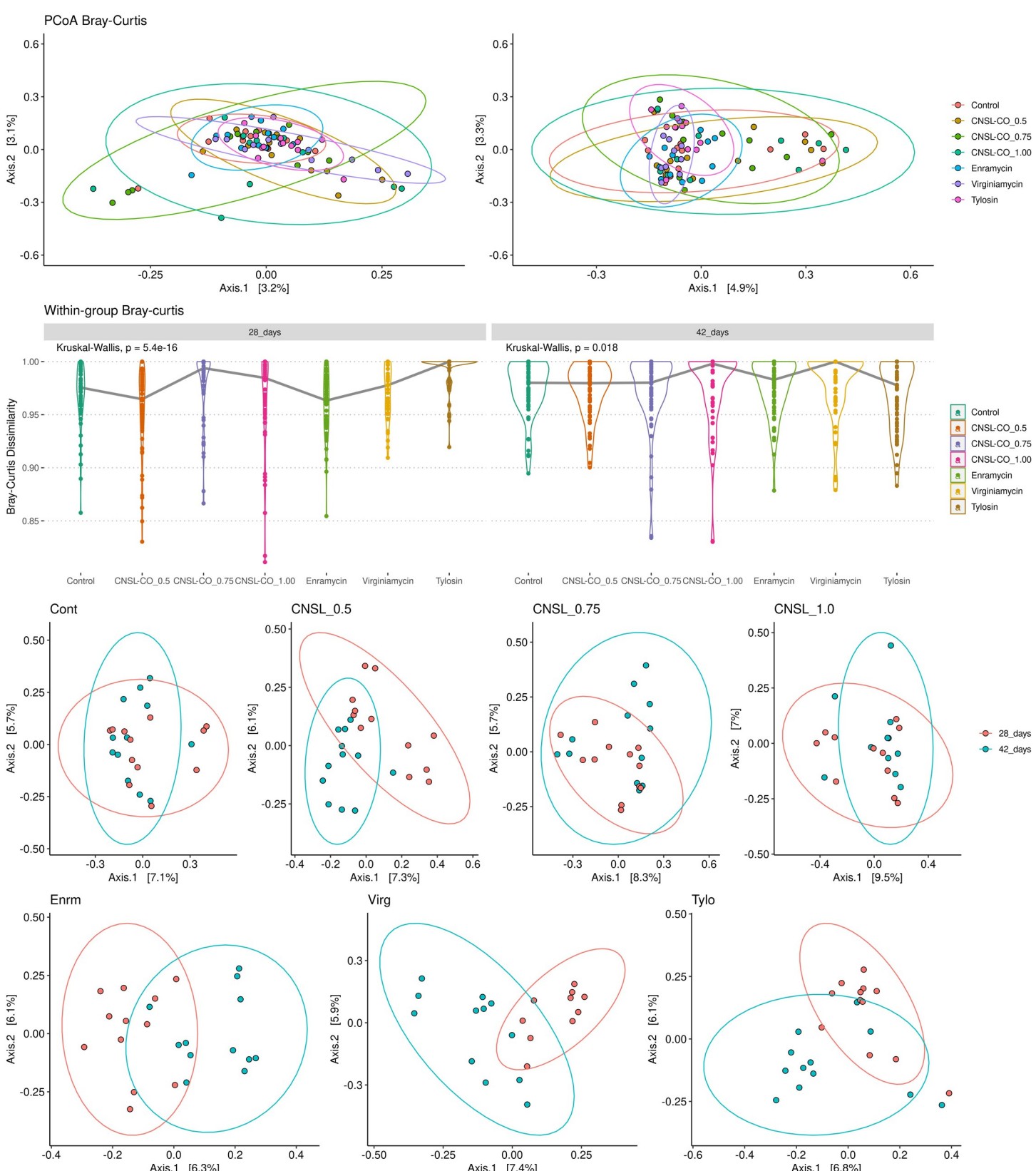

**Fig 4. Beta diversity based on Bray–Curtis dissimilarity of broilers cecum samples at 28 and 42 d of trial.** The study tested seven feed additives: enramycin (8 ppm), virginiamycin (16.5 ppm), tylosin (55 ppm), CNSL-castor oil (CNSL-CO) at different doses (0.5, 0.75, and 1.00 kg/t), and the control diet (without additives). (A) Comparison between all treatments at each time point, with (top) the principal coordinate analysis (PCoA) plot and (bottom) the differences in within-group Bray–Curtis distances. (B) Comparison for each additive between time points.

abundance > 1% (S3 Table and Fig 5B). The phylum Firmicutes was the most abundant, regardless of treatment and age, with values between 84.33% and 95.16%. Ruminococcaceae, Lachnospiraceae, and Lactobacillaceae families were predominant in all groups (>15%), mainly at 42 d.

## Discussion

In this study, we compared the effects of increasing CNSL–castor oil levels and growth-promoting antibiotics tylosin, enramycin, and virginiamycin associated with the anticoccidials semduramicin + nicarbazin and sodium monensin in the initial phase (0 to 28 d) broilers challenge with coccidiosis. At 21 d of age, the animals that received the antibiotic tylosin showed comparatively better performance. However, at 42 d, there was no difference in performance among the antibiotic treatments. The use of CNSL–castor oil 0.75 and 1 kg/t treatments did not differ from treatments with enramycin and virginiamycin, although these CNSL–castor oil groups did present better feed conversion."

Antibiotics and anticoccidials (chemicals and ionophores) are widely used together, alone, or in combination in coccidiosis prevention programs. When clinical signs become apparent, it is too late to prevent pathological consequences and decreased performance caused by parasite infection [17]. Tylosin is an antibiotic with beneficial effects on broiler performance. Hung

**Table 2. Adonis test result of the beta diversity based on Bray–Curtis dissimilarity of broilers cecum samples at 28 and 42 d of trial with seven feed additives: Enramycin (8 ppm), virginiamycin (16.5 ppm), tylosin (55 ppm), CNSL-castor oil (CNSL-CO) at different doses (0.5, 0.75, and 1.00 kg/t), and the control diet (without additives).**

| Variable | F.model | R2 | p-value |
|---|---|---|---|
| **Additive + Time** | | | |
| Additive | 1.1093 | 0.0400 | 0.002 |
| Time | 2.6391 | 0.0159 | 0.001 |
| **Time** | | | |
| Time 28D | 1.0361 | 0.0775 | 0.155 |
| Time 42D | 1.0947 | 0.0786 | 0.001 |
| **Additive/Time** | | | |
| Control | 1.098 | 0.0475 | 0.065 |
| CNSL-CO_0.5 kg/t | 1.4735 | 0.0628 | 0.001 |
| CNSL-CO_0.75 kg/t | 1.0672 | 0.0463 | 0.164 |
| CNSL-CO_1.0 kg/t | 1.1944 | 0.0538 | 0.057 |
| Enramycin | 1.3004 | 0.0558 | 0.002 |
| Virginiamycin | 1.3845 | 0.0647 | 0.001 |
| Tylosin | 1.2325 | 0.0531 | 0.002 |
| **Time 42/Control** | | | |
| CNSL-CO_0.5 kg/t | 1.0386 | 0.0451 | 0.248 |
| CNSL-CO_0.75 kg/t | 1.0012 | 0.0435 | 0.432 |
| CNSL-CO_1.0 kg/t | 1.0774 | 0.0467 | 0.182 |
| Enramycin | 1.0789 | 0.0468 | 0.079 |
| Virginiamycin | 1.1194 | 0.0484 | 0.045 |
| Tylosin | 1.0959 | 0.0475 | 0.080 |

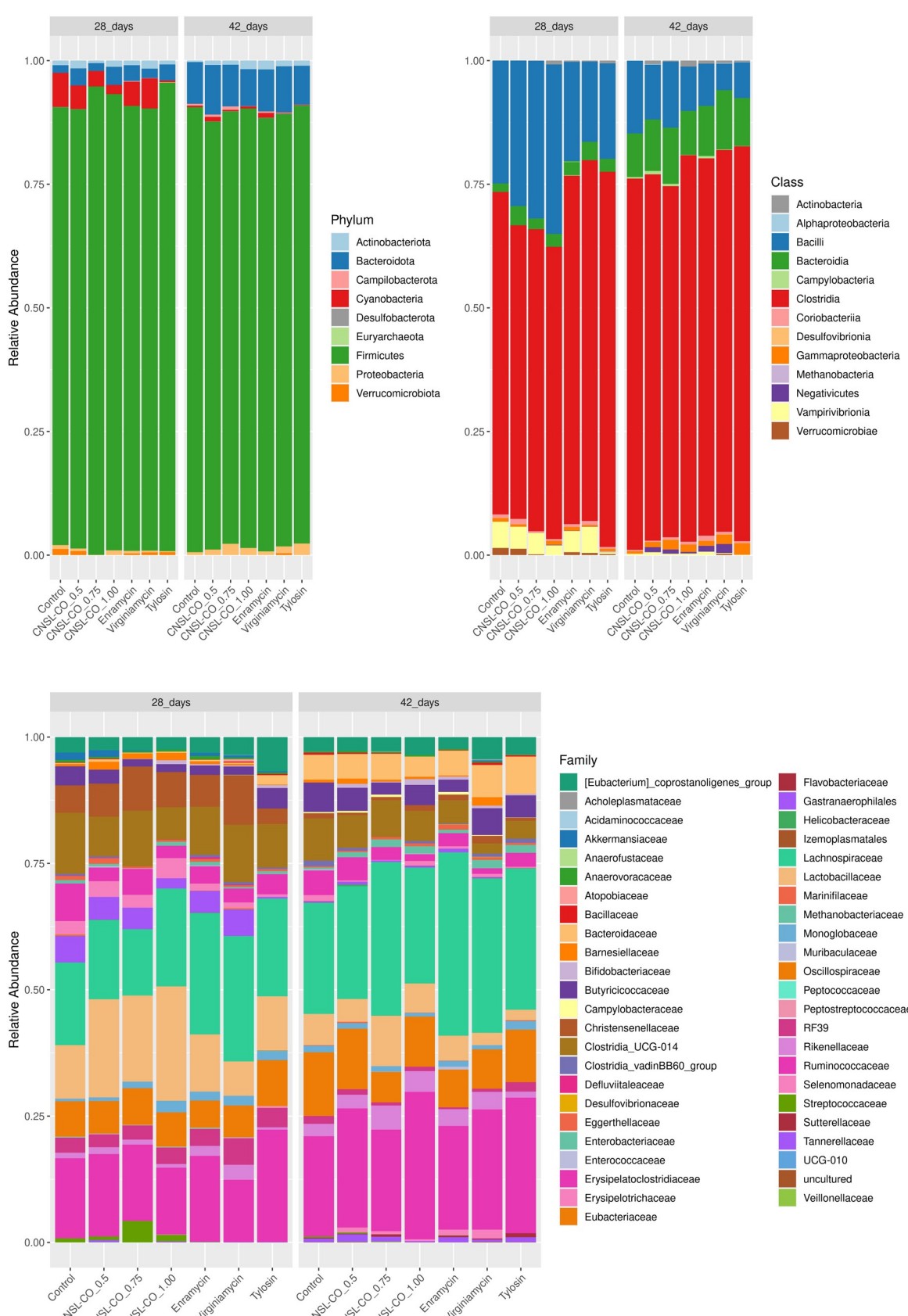

**Fig 5. Relative abundance of broilers cecum samples at 28 and 42 d of trial.** The study tested seven feed additives: enramycin (8 ppm), virginiamycin (16.5 ppm), tylosin (55 ppm), CNSL–castor oil (CNSL-CO) at different doses (0.5, 0.75, and 1.00 kg/t), and the control diet (without additives). Relative abundances are presented in percentage (%) of taxa at (A) Phylum and Class, and (B) Family levels.

et al. [18] observed that the use of 55 mg/kg tylosin provided beneficial effects on digestion and intestinal integrity, increasing disaccharidase activity and maintaining intestinal permeability, consequently increasing performance.

Although the advantages of using antibiotics as growth promoters have been extensively studied and reviewed, consumer pressure to reduce the use of antibiotics and AGP for animal production is high, as these treatments have been linked to the rise in antibiotic-resistant bacteria for humans [19]. Cardinal et al. [20] concluded that AGP withdrawal has a negative impact, mainly in the initial stages of broiler performance. Corroborating the results of this study, animals that received tylosin showed better performance in the first phase and remained until the end of the experiment. Brazil banned the use of tylosin and other antibiotics, such as tiamulin and lincomycin, in 2020 [21], indicating the importance of studies comparing alternative substances.

However, ionophores are considered antibiotics of non-medical importance for human health [21], and in animal production, they are the primary choice to control coccidiosis. One important factor to consider is that they do not entirely suppress parasite development, thus allowing the development of host immunity after initial exposure [17, 22]. Parent et al. [23] observed that programs that used antibiotics presented similar performance to those that exclusively used ionophores. However, cases of parasite resistance to ionophores have been reported owing to the use of these substances for an extended period [22].

Studies with different phytogenics have shown promising results for broiler performance when the birds are challenged by coccidiosis [24–26]. Moraes et al. [8] compared the effect of CNSL–Castor oil with that of monensin in broilers challenged with coccidiosis and found similar results between the functional oil and the ionophore. The mechanisms of action of CNSL–castor oil have not been fully elucidated. However, in the face of the challenge caused by coccidiosis, CNSL–Castor oil acts as an immunomodulator [27] and modulator of the intestinal microbiota [28]. CNSL–Castor oil helps maintain the animal's immune systems, eliminate parasites, maintain intestinal balance, and improve animal performance [8, 28, 29].

This study observed a higher lesion score for *E. acervulina* and *E. tenella* in the control group compared with that in other treatments, demonstrating the beneficial action of CNSL–Castor oil during coccidiosis challenge (21 d), similar to the effect of antibiotics. However, the same pattern was not observed at 28 d, where there was no difference in the intestinal lesion score regardless of treatment. Alternatives to antibiotics can become ineffective in an environment with compromised biosafety conditions and management errors, which can be a limiting factor for the exclusive use of alternatives to antibiotics [30, 31]. In the field, combining alternative products with anticoccidials is standard practice. In this study, the use of anticoccidials in the initial phase, followed by CNSL–Castor oil, resulted in a similar performance to that of antibiotics. This study used lower levels of CNSL–Castor oil (0.75 and 1.0 kg/t) than previous studies that have demonstrated beneficial effects using a dose of 1.5 kg/t.

Invasion by coccidian parasites into intestinal cells disturbs microbiota homeostasis in the healthy intestine [32]. Changes in the intestinal environment caused by these parasites include changes in nutrient absorption and digestibility, increased mucogenesis, membrane permeability, nutrient availability, and the proliferation of pathogenic bacteria [33].

The microbial richness and diversity of the gut are closely related to the health of broilers [34]. Coccidiosis infection alters the intestinal microbiota profile, reducing the diversity of the

intestinal environment, especially in the cecum, which has the greatest biodiversity [9, 35]. However, microbiota modulators can reduce this impact, making the animals resilient during coccidiosis [9, 31, 36, 37]. In this study, when animals were evaluated at 28 and 42 d of age, it was observed that even after treatment with antibiotics, there was an increase in microbial diversity, except in groups treated with enramycin or 1 kg/t CNSL–castor oil. In a separate study, Vieira et al. [9] demonstrated that the use of 1.5 kg/t CNSL–castor oil maintained similar diversity at 7 and 14 d after the coccidiosis challenge, in contrast to the use of 100 ppm of sodium monensin.

As in other studies, Firmicutes was the most abundant phylum in the cecum, but its abundance decreased with the age of the chickens [38]. Firmicutes, as it comprises a large proportion of commensal bacteria, is associated with the efficiency of energy capture, and consequently, with better product performance [39]. In this study, at 28 d, except for virginiamycin, all treatments showed a greater abundance of this phylum than the control. Corroborating the results of this study, Vieira et al. [9], working with CNSL–castor oil alone, also observed an increase in Firmicutes in broilers challenged with coccidiosis, which was not observed when only monensin was added. Using this same product, an increase in Firmicutes was observed in newly weaned piglets; this a phase where there is a disturbance in the balance of the intestinal microbiota [9, 40].

The phylum Actinobacteria also increased in all treatments compared with that in the control. Bacteria representing this phylum are aerobic and represent a small percentage of the intestinal microbiota. However, Actinobacteria can maintain intestinal homeostasis and are negatively correlated with Proteobacteria, including pathogenic species such as *Salmonella* and *Escherichia*. Proteobacteria have been associated with poor performance in broilers [41]. This fact was also observed in this study, as there was a reduction in Proteobacteria at 28 d—that is, 14 d after the challenge—compared with that in the control treatment. Previous studies have demonstrated that CNSL–castor oil reduces the abundance of Proteobacteria at times of stress in different animals, whether under coccidiosis in chickens [9] or at weaning in pigs [28]. Conversely, [42] it has been observed that monensin supplementation produced a greater relative abundance of gram-negative bacteria of the phylum Proteobacteria and class Clostridia. Danzeisen et al. [43] observed that the combination of monensin with virginiamycin or tylosin increased the presence of this phylum, especially *E. coli*.

Among these families, two considered pathogenic are important to highlight. Streptococcaceae, facultatively anaerobic bacteria, produce lactic acid and are characterized as an indication of dysbiosis after infection by *Eimeria tenella* [44]. All treatments in the current study reduced the abundance of this family compared with the levels seen in the control. The Enterobacteriaceae family, also facultative anaerobes and lactic acid producers, presented lower abundance in the CNSL–Castor oil, tylosin, and virginiamycin treatments relative to that in the control. In this family, the main pathogenic bacteria are related to the most significant moments of stress in broiler chickens [9, 45].

The families Erysipelotrichaceae, Lachnospiraceae, Rikenellaceae, and Ruminococcaceae are anaerobic bacteria associated with the fermentation of structural carbohydrates and the production of short-chain fatty acids [44, 46]. The families Lachnospiraceae and Ruminococcaceae showed greater abundance in treatments with tylosin and enramycin when compared with the control. All CNSL–Castor treatments decreased the abundance of the Ruminococcaceae, Lachnospiraceae, and Rikenellaceae families at 28 d compared with those in the control.

Previous studies have shown that different AGPs can enrich the cecum with butyrate-producing bacteria [47, 48]. Robinson et al. [49] observed that tylosin and enramycin increased the abundance of Ruminococcacea, whereas salinomycin and monensin reduced the abundance of this family. In this study, after the removal of the ionophores, at 42 d, there was an

increase in the abundance of Ruminococcaceae, Lachnospiraceae, and Rikenellaceae in all CNSL–castor oil treatments.

The Erysipelotrichaceae family showed greater abundance under the 0.75 and 1 kg/t CNSL–castor oil treatments in both periods. Vieira et al. [9] observed that 1.5 kg/t CNSL–castor oil treatment for chickens challenged with coccidiosis increased the abundance of bacteria associated with the production of short-chain fatty acids, such as Erysipelotrichaceae, compared with treatment with monensin. This corroborates the results of this study, which found that low levels of CNSL–castor oil, when associated with anticoccidials, reduced the abundance of this family when evaluated at 28 d. However, at 42 d, in the absence of anticoccidials, there was an increase in the abundance of this family.

With the exception of the virginiamycin treatment, all treatments showed a relative increase in the abundance of the Lactobacillaceae family at 28 d compared with that in the control. However, at 42 d, all treatments showed a reduction in the relative abundance of this family. Members of this family can survive in both aerobic and anaerobic environments. During challenge by *Eimeria* spp., this family proliferates and contributes positively to weight gain, reduces injury scores, and increases mucosal integrity [39, 46]. Furthermore, many *Lactobacillus* spp. act on the immune system by stimulating immune cells to release pro-inflammatory cytokines, such as tumor necrosis factor-alpha (TNF-α), interferon-gamma (IFN-γ), and interleukin-12 (IL-12), triggering an immunomodulatory response [24, 50].

The use of different subtherapeutic doses of growth-promoting antibiotics has different effects on bacterial adhesion in the cecum [51], especially when added with anticoccidials. It is noteworthy that the microbiota profile that leads to overall improved performance has not been established yet, as different intestinal microbiota compositions can result in similar growth performances [2].

## Conclusions

Tylosin showed better performance indices in chickens challenged with coccidiosis. The CNSL–castor oil levels of 0.75 and 1 kg/t were more effective than a level of 0.50 kg/t, presenting similar results to enramycin and virginiamycin. Furthermore, CNSL–castor oil acted as a modulator of intestinal microbiota, reducing the abundance of pathogenic bacteria.

## Supporting information

**S1 Fig.** Lesion score in coccidiosis challenged broilers at 21 (A) and 28 (B) days of age—7 and 14 d after challenge, respectively. CNSL–Castor (US Patent N. 8377,485; Oligo Basics Ind. Ltda., Cascavel, Parana, Brazil). Means with different letters differ statically as per the Student–Newman–Keuls method.
(PNG)

**S2 Fig.** A, B. Comparison of alpha diversity for each additive between time points using CHAO -1 (A) and Simpson (B) index of broilers cecum samples at 28 and 42 d of trial. The study tested seven feed additives: enramycin (8 ppm), virginiamycin (16.5 ppm), tylosin (55 ppm), CNSL–castor oil (CNSL-CO) in different doses (0.5, 0.75, and 1.00 kg/t), and the control diet (without additives).
(PNG)

**S3 Fig.** A, B. Comparison of alpha diversity at each time point using CHAO -1 (A) and Simpson (B) index of broilers cecum samples at 28 and 42 d of trial The study tested seven feed additives: enramycin (8 ppm), virginiamycin (16.5 ppm), tylosin (55 ppm), CNSL–castor oil (CNSL-CO) at different doses (0.5, 0.75, and 1.00 kg/t), and the control diet (without

additives).
(PNG)

**S1 Table. Ingredient formulae and chemical composition of experimental diets according to the rearing period.**
(DOCX)

**S2 Table. Number of reads that passed through each step of quality control for the experiment testing seven feed additives: Enramycin (8 ppm), virginiamycin (16.5 ppm), tylosin (55 ppm), CNSL–castor oil (CNSL-CO) at different doses (0.5, 0.75, and 1.00 kg/t), and the control diet (without additives) of broilers cecum samples at 28 and 42 d of trial with 12 replicates.**
(XLSX)

**S3 Table. Relative abundance at phylum, family, and genera levels present in the experiment.** The experiment tested seven feed additives: enramycin (8 ppm), virginiamycin (16.5 ppm), tylosin (55 ppm), CNSL–castor oil (CNSL-CO) at different doses (0.5, 0.75, and 1.00 kg/t), and the control diet (without additives) of broilers cecum samples at 28 and 42 d of trial with 12 replicates.
(XLSX)

## Author Contributions

**Conceptualization:** Lucélia Hauptli, Douglas Haese, Carolina D'ávila Pozzatti, Priscila de Oliveira Moraes.

**Formal analysis:** Tatiany Aparecida Teixeira Soratto, Vilmar Benetti Filho, Carolina D'ávila Pozzatti, Priscila de Oliveira Moraes.

**Investigation:** Pedro Torres, Tatiany Aparecida Teixeira Soratto, Douglas Haese, Carolina D'ávila Pozzatti.

**Methodology:** Paula Gabriela da Silva Pires, Tatiany Aparecida Teixeira Soratto, Vilmar Benetti Filho, Lucélia Hauptli, Glauber Wagner, Douglas Haese, Carolina D'ávila Pozzatti.

**Project administration:** Douglas Haese, Carolina D'ávila Pozzatti.

**Writing – original draft:** Paula Gabriela da Silva Pires, Pedro Torres, Tatiany Aparecida Teixeira Soratto, Vilmar Benetti Filho, Lucélia Hauptli, Priscila de Oliveira Moraes.

**Writing – review & editing:** Glauber Wagner, Priscila de Oliveira Moraes.

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
