## [Decision Letter · Decision Letter 0]

16 May 2022

PONE-D-22-04312Comparison of the use of a blend of functional oils with antibiotics associated with anticoccidials on the performance and microbiota of broiler chickens challenged by coccidiosisPLOS ONE

Dear Dr. Moraes,

Thank you for submitting your manuscript to PLOS ONE. After careful consideration, we feel that it has merit but does not fully meet PLOS ONE’s publication criteria as it currently stands. Therefore, we invite you to submit a revised version of the manuscript that addresses the points raised during the review process.

Both reviewers consider the data interesting and sufficient for publication in Plos One.  However, both reviewers believe that the majority of issues that need to be addressed a editorial in nature and, in fact, may require a review by a native English speaker.  Please direct all of your revisions to directly to the comments/suggestions made by the reviewers on the content of the revision. 

We look forward to receiving your revised manuscript.

Kind regards,

Michael H. Kogut, Ph.D.

Academic Editor

PLOS ONE

Journal Requirements:

3. Please include your tables as part of your main manuscript and remove the individual files. Please note that supplementary tables (should remain/ be uploaded) as separate "supporting information" files

Reviewers' comments:

Reviewer's Responses to Questions

**Comments to the Author**

1. Is the manuscript technically sound, and do the data support the conclusions?

Reviewer #1: Yes

Reviewer #2: Partly

2. Has the statistical analysis been performed appropriately and rigorously? 

Reviewer #1: Yes

Reviewer #2: Yes

3. Have the authors made all data underlying the findings in their manuscript fully available?

Reviewer #1: Yes

Reviewer #2: Yes

4. Is the manuscript presented in an intelligible fashion and written in standard English?

Reviewer #1: Yes

Reviewer #2: No

5. Review Comments to the Author

Reviewer #1: Identifying effective antibiotics alternatives is important for the poultry industry. This manuscript provides some insight on beneficial roles of these alternatives on growth performance and gut microbiome changes.

There are some minor comments:

1. Line 26: Add "A total of"

2. Line 30: Be consistent throughout the mansucript: (0.5 kg / t; 0.75 kg / t; 1.00 kg / t) vs. (0.5 kg/t; 0.75 kg/t; 1.00kg/t)

3. Line 119: the Brazilian Tables of Pigs and Swine must be incorrect. This is a poultry experiment. Use a poultry requirement table.

4. Table1 & 2 should stand alone. Add more notes/legends: define CNSL-CO; N = ???

5. Figure 3: define P values: *, **, ***, NS

Reviewer #2: Dear authors, the work is interesting, the number of replications is very good, and the results help to improve the understanding of the challenges imposed by coccidiosis and the products used in the market to combat it. However, there are many details that need to be fine-tuned to make the work clear to readers. Important information is missing from the summary and there are problems throughout the text that make it very difficult to read. I put a lot of questions in the list that I made throughout the reading. In addition, a review by native English speakers will be critical for future publication, as the text has many problems. There are sentences in which the problem is not the language, but the structure of the sentence itself, which is incorrect and leads to confusion in interpretation.

6. PLOS authors have the option to publish the peer review history of their article (what does this mean?). If published, this will include your full peer review and any attached files.

Reviewer #1: No

Reviewer #2: No

---

## [Author Response · Author response to Decision Letter 0]

6 Jun 2022

Response letter to reviewers

Dear reviewer, thank you for taking your time and for adding knowledge to our survey with your experience. We hope that the answers to your questions are satisfactory, if you still have any questions, we are at your disposal for any clarification. The text was sent to Editage company for English correction, the changes made by the Editage company are marked by the word corrector and those requested by you are marked in yellow.

Review 1

Reviewer #1: Identifying effective antibiotics alternatives is important for the poultry industry. This manuscript provides some insight on beneficial roles of these alternatives on growth performance and gut microbiome changes.

There are some minor comments:

1. Line 26: Add "A total of" – 

The term has been added to the text. Thanks

2. Line 30: Be consistent throughout the mansucript: (0.5 kg / t; 0.75 kg / t; 1.00 kg / t) vs. (0.5 kg/t; 0.75 kg/t; 1.00kg/t). 

Units have been standardized throughout the text.

3. Line 119: the Brazilian Tables of Pigs and Swine must be incorrect. This is a poultry experiment. Use a poultry requirement table.

The term has been changed. Thanks

4. Table1 & 2 should stand alone. Add more notes/legends: define CNSL-CO; N = ???

The number has been added

5. Figure 3: define P values: *, **, ***, NS

The values were added in the figure legend.

Reviewer 2

 TITLE: Comparison of the use of a blend of functional oils with antibiotics associated with anticoccidials on the performance and microbiota of broiler chickens challenged by coccidiosis 

 I´d like to understand where authors saw an association of antibiotics and anticoccidials, because I can´t see it. The antibiotics were used in almost all the period alone. One in one week monensin was used together with the antibiotics . So, what do you mean with this “association”? For e associated means together, with, and not after.

Dear reviewer, thank you for the opportunity to explain this part better. All treatments contained semduramycin + nicarbazin (500g/t - Aviax® Plus) from 0 to 28 days of age and sodium monensin (100 ppm - Elanco) from 29 to 35 days of age. Only in the last 7 days were only the functional oils or antibiotics. For this reason we understand that it was an association, since semduramycin, nicarbazine and monensin sodium are anticoccidial. In the abstract this information was corrected, in the material and methods it is possible to check between lines 123 and 126. In supplemental table 1 these dosages are described in the formulation.

Abstract e M&M it was not 2520 divided in 12 pens, but in 84 pens

Thanks for the correction, it really was a mistake, it was fixed.

Line 33 28 D). The challenge was carried out by inoculating broiler chickens with

34 sporulated Eimeria tenella, Eimeria acervulina, and Eimeria maxima oocysts via35 oral gavage. In addition to performance parameters, intestinal contents were36 collected at 28 and 42 days of age for microbiota analysis by sequencing 16s37 rRNA in regions V3 and V4 using the Illumina MiSeq platform

It is lacking- When, in what age was done the inoculation?

The challenge was carried out at 14 days of life, it was a mistake not to put it in the abstract, it has already been corrected

Line 31-All treatments received semduramicin + nicarbazin (500g/t) and sodium monensin (100ppm) in the initial live phase (0 to 28 D). In the M&M you say that monensin was from 29 days until 35 (Line 122).

Thank you, this information has been corrected in the abstract.

“There was increasing diversity from 28 D to 42 D using the three indices “– you did not told us what indices are you talking about…

Thank you, this information has been corrected in the abstract.

“There was increasing diversity from 28 D to 42 D using the three indices, which were significant only for the Simpson index and showed higher time dominance at 42 days”- in what treatment? Only due to an age influence? In other words, why there was an increase in diversity? 

This excerpt has been rewritten in the abstract for better understanding. As abstract is succinct, we chose to write only the age effect.

In the item "results" it is possible to observe the presentation of the result by more detailed treatment, between lines 244 to 251. In the item "discussion" between lines 391 - 401 we discussed the fact that coccidiosis reduces microbial diversity, and possibly this affected the diversity at 28 days leading to greater diversity at 42 days.

Line 39 “In the evaluation at 21 days, the broilers that received tylosin had…! Why a 21 day response if you´ve changed diet only at 28 days? I onçy understood this response nowing challenge was at 14 days. So, rewrite saying that “ after one week of challenge or something like this…”

The text has been changed. 

“at 42 days, it had the lowest”. It had the lowest what??

Thanks for the opportunity to correct, firmicutes was the biggest phylum at both ages.

“Tylosin showed better performance indices of chickens challenged by coccidiosis.” You did not tell us if there was some treatment WITHOUT challenge. How can we understand?

All the chickens were challenged, the phrase really became redundant. The text has been corrected

” . On the other hand, the antibiotics also increased the abundance of bacteria producing short-chain fatty acids from 28 days” – what do you mean with from 28 days? AFTER 28 days? Or from 28 up to 42 days.

The corrector term would really be "after"

Introduction

Line 67 “significant increase in its proliferation in the small intestine and it can cause necrotic enteritis [2,3]

The text has been changed. 

Line 74 - and direct inflammatory [5]. How AGP causes inflamation? It is the contrary.

The text has been changed. It's really the opposite

M&M

Line 110 - the birds could have euthanized? 111 was no specific euthanasia situation : the coccidiosis infection responded to the…

The text has been changed.

Line 13. The experimental period lasted 42 days and broilers were Weekly weighed, measured feed intake was measured, and calculated feed

Conversion were calculated [11] Please change place (from Line 124 to here). The information is loose up there.

The text has been changed.

Line 188 The performance responses separated according to the periods 1 to 21 ( no :) 

The text has been changed.

191 presented more significant weight gain ?

... presented greater weight gain , with a statistical significance (is that you mean???)

The text has been changed.

Line 102- while the other treatments demonstrated an intermediate 193 behavior, and consequently, had higher live weight and better FCR (p < 0.05). You wrote that the OTHER TREATMENTS had better FCR an I think you wanna mean that tylosin treatments had netter FCR. Rewrite the sentence, please.

The text has been changed.

194 The control group showed worse feed conversion (P < 0.05). Rewrite please. Be more direct. 

The text has been changed.

Line 200 –“The following period was completed 14 days after the challenge by Eimeria spp (14 to 28 D)”… you´ve told about 3 periods (The performance responses separated according to the periods- 1 to 21 days, 21 to 28 days, 28 to 42 days, and a total period of 1 to 42 days.) . What following period? Not even in Table 1 is this period.

The text has been changed.

219 0.05). Consequently, the pattern for live weight at 42 days was identical to weight gain (identical are the same numbers)

The text has been changed.

 Line 226 In the first week after infection, the broilers submitted to the control treatment had a higher lesion score for E. acervulina and E. tenella when compared to the other treatments (p < 0.05). Results not shown???

The information is presented in supplementary figure 1. This information is relevant because it demonstrates that there was a greater impact of coccidiosis in the treatment without antimicrobial and without the phytogenic. However, in order not to overload the article with images, we decided to leave it as a supplementary figure.

246 The same trend??? occurred for all observed indices (Fig. S2 A and SB)

The text has been changed.

260 Comparing the different additives at each time point, we observed

261 statistically significant differences between them (Kruskal-Wallis, P < 0.05). Each treatment was pairwise compared using the Wilcoxon test corrected by the FDR263 (False Discovery Rate). This is not results. It is M&M

At 28 days, there were no significant differences compared to the control (Fig. 3). At 42 days, the control differed from treatments with virginiamycin, CNSL–castor oil 1.00kg/t, and enramycin (Fig. S3A e S3B). I did not find Fig S3A and Fig S3B. 

Why the comparison towards control treatment? In my opinion, or you explain all differences ( a lot, may be) or just say about the existence of differences,

The text has been changed. The images are in the system in the journal, I don't understand why it didn't arrive for your review.

We saw that the control is our reference point, we look for additives to improve the intestinal health and the performance of the animals and the control is our parameter, because if it is not better than the control, we might have to rethink the test additive.

277 samples were scattered with overlapping ellipses (Fig. 4A). At time 42,? At 42 days of age? 280 treatments at each studied time, 281 are not similar. Comparing each treatment at different times (periods?// days??) times is not an appropriate term 283 time 28 days to time 42 days (Adonis, p < 0.05) (Fig. 4B and Table 2). Time again

We do not fully understand its placement, but we have organized the text by replacing the word time e and placing the day of the analysis in figure 4A.

328 and 1kg/t did not differ from the treatment with enramycin and virginiamycin, with a better feed conversion when compared to the control treatment. Here you are talking about the comparison between enramycin , virginiamycin x castor oil. Leave control out this comparison.

The sentence has been changed. Thanks.

Line 348- On the other hand, ionophores are considered antibiotics of non-medical importance for human health [21] (.periodo) In animal production,end point…

The end point has been added. Thanks.

353 used antibiotics had similar zootechnical performances… It is not an usual word in English

The term has been deleted. Thanks.

Line 354 - …However, cases of parasite resistance have already been

355 reported due to the use of these substances for an extended period [22]. What substances? Antibiotics or ionophores?? Not clear. Use the expressions “the last, the former”

The sentence has been updated to make it easier for the reader to understand.

366 This study observed that even using anticoccidials, there was a higher

367 lesion score for E. acervulina and E. tenella for the control compared to the other368 treatments, which demonstrates beneficial action of CNSL-Castor oil during369 coccidiosis challenge, similar to the effect of antibiotics. Wrong sentence. What do you wanna say? If anticocidials were totatlly efficient we should expect NO lesion score. So, even using anticoccidial it did not work perfectly. And after you continuous… However, alternatives to 370 antibiotics can become ineffective in an environment with fewer biosafety. However? But it did not work perfectly. Why however? 

We agree. The section was changed for a better understanding of the text. Thanks

 373 In this study, added the use of anticoccidials in the initial phase FOLLOWED BY CNSL-Castor oil allowed a similar performance to antibiotics. 

The text has to be clear that was a sequence not an addition, not an association.

The sentence was changed as per the suggestion. Thanks

377- This study used lower levels of CNSL-Castor oil (0.75 and 1.0 kg/t)

376 compared to previous studies, evaluating its isolated effect, demonstrated its beneficial effect with 1.5 kg/t [8,29]. 

English is not correct. Rewrite the sentence

The sentence was rewritten.

377 found no synergistic effect between oregano essential oil and monensin at low doses. The combination of the two additives showed similar effects to the substances used separately in broilers challenged by coccidiosis. Why do youpoint this? In your artile there was no associations to see synergistic effect. This citation is lost in the contend. 377 found no synergistic effect between oregano essential oil and monensin at low doses. The combination of the two additives showed similar effects to the substances used separately in broilers challenged by coccidiosis. This part of the article is discussion, not introduction.

We are in agreement. The sentence has been removed. Thanks

386 The microbial richness and diversity of the gut are closely related to broiler ´s health [34].

The sentence has been changed as suggested. Thanks.

391- In this study, when evaluating at 28 and 42 days of age, it was observed that even within treatments containing antibiotics, there was a change in microbial diversity, except for enramycin and CNSL-castor oil 1kg/t. How is this change? Incresed?? Decreased?

There was an increased there was a change in microbial diversity, the sentence was rewritten.

Line 394- 1.5kh/t of CNSL What unit is that?

The unit has been modified. A typing error has occurred. Thanks

411- However, it has been able to maintain intestinal homeostasis and negatively correlated with Proteobacteria, which are mostly facultative or obligate anaerobes with flagella or with the ability to move by gliding, including pathogenic species such as Salmonella and Escherichia. In broiler chickens, it has been associated with poor performance [41].

Sentences difficult to understand. Rewrite, please. Readers cannot understand if Proteobacteria is good or not

The sentence has been rewritten to make it easier for readers to understand.

439 showed greater abundance for treatments with Tylosin and Enramycin (when) compared to the control - 444 butyrate-producing bacteria [47,48]. Robinson et al. [49]) used tylosin and 445 observed that Tylosin and Enramycin increased. Tylosin with capital letter or not?

The word has been standardized throughout the text. Thanks for the observation.

449- The Erysipelotrichaceae family showed greater abundance at (wrong preposition) the 0.75 and

The preposition was changed.

454- Corroborating the results of this study, which found 455 that lower levels of CNSL-castor oil were associated with anticoccidials and 456 increasing abundance at 42 days in their absence. The structure of the sentence is wrong. Do you mean the study of Vieira et al corroborate the results of the present study? And what do you intend to say about … ” lower levels of CNSL-castor oil were associated with anticoccidials and increasing abundance at 42 days in their absence.”? How low levels of castor oil are associated with anb? They increase the abundance of what? 458- abundance increase? How you know it is an increase since you did not measure BEFORE 28 days?

The sentence has been rewritten to make it easier for readers to understand.

468 The use of different subtherapeutic doses of growth-promoting antibiotics 469 has different effects on bacterial adhesion of the cecum, (coma) [51] especially when470 added to the use of anticoccidials.

The comma has been added.

472 microbiota compositions can result in similar growth performances, as seen in this research [2]. I don´t think this is a corret manner to cite the article

The citation was revised. Thanks. 

Conclusion On the other hand, the antibiotics increased

481 the abundance of bacteria producing short-chain fatty acids at 28 days. For me, this is not clear in you discussion and it shows up in the abstract an in the conclusion. Or you emphasize in the discussion and in the results or you take it out.

The sentence was withdrawn.

---

## [Editor Report · Decision Letter 1]

9 Jun 2022

Comparison of functional-oil blend and anticoccidial antibiotics effects on performance and microbiota of broiler chickens challenged by coccidiosis

PONE-D-22-04312R1

Dear Dr. Moraes,

We’re pleased to inform you that your manuscript has been judged scientifically suitable for publication and will be formally accepted for publication once it meets all outstanding technical requirements.

Kind regards,

Michael H. Kogut, Ph.D.

Academic Editor

PLOS ONE
---

## [Editor Report · Acceptance letter]

14 Jun 2022

PONE-D-22-04312R1 

Comparison of functional-oil blend and anticoccidial antibiotics effects on performance and microbiota of broiler chickens challenged by coccidiosis 

Dear Dr. Moraes:

I'm pleased to inform you that your manuscript has been deemed suitable for publication in PLOS ONE. Congratulations! Your manuscript is now with our production department. 

Kind regards, 

on behalf of

Dr. Michael H. Kogut 

Academic Editor

PLOS ONE